

# High-resolution temperature profiles (HRTP) retrieved from bi-chromatic stellar scintillation measurements by GOMOS/Envisat

Viktoria F. Sofieva[1], Francis Dalaudier[2], Alain Hauchecorne[2], and Valery Kan[3]

[1] Finnish Meteorological Institute, Helsinki, Finland
[2] Université Versailles St-Quentin, Sorbonne Université, CNRS/INSU, LATMOS-IPSL, Guyancourt, France
[3] A.M. Obukhov Institute of Atmospheric Physics, Moscow, Russia

*Correspondence to*: V.F. Sofieva (viktoria.sofieva@fmi.fi)

## Abstract

In this paper, we describe the inversion algorithm for retrievals of high vertical resolution temperature profiles using bi-chromatic stellar scintillation measurements in the occultation geometry. This retrieval algorithm has been improved with respect to nominal ESA processing and applied to the measurements by Global Ozone Monitoring by Occultation of Stars (GOMOS) operated on board Envisat in 2002-2012. The retrieval method exploits the chromatic refraction in the Earth's atmosphere. The bi-chromatic scintillations allow the determination of the refractive angle, which is proportional to the time

delay between the photometer signals. The paper discusses the basic principle and detailed inversion algorithm for reconstruction of high resolution density, pressure and temperature profiles (HRTP) in the stratosphere from scintillation measurements. The HRTP profiles are retrieved with very good vertical resolution of ~200 m and high accuracy of ~1-3 K for altitudes of 15-32 km and with a global coverage. The best accuracy is achieved in in-orbital-plane occultations, and the accuracy weakly depends on star brightness. The whole GOMOS dataset has been processed with the improved HRTP

inversion algorithm using the FMI's Scientific Processor; and the dataset (HRTP FSP v1) is in open access.

The validation of small-scale fluctuations in the retrieved HRTP profiles is performed via comparison of vertical wavenumber spectra of temperature fluctuations in HRTP and in collocated radiosonde data. We found that the spectral features of temperature fluctuations are very similar in HRTP and collocated radiosonde temperature profiles.

HRTP can be assimilated into atmospheric models, used in studies of stratospheric clouds and in analysis of internal

gravity waves activity. As an example of geophysical applications, gravity wave potential energy has been estimated using the HRTP dataset. The obtained spatio-temporal distributions of gravity wave energy are in good agreement with the previous analyses using other measurements.

## 1    Introduction

This paper is dedicated to the description of a unique method for high-resolution temperature and density profiling using bi-

chromatic satellite stellar scintillation measurements and to assessment of the retrieved temperature profiles. The bi-



chromatic stellar scintillation  measurements were performed by two fast photometers at different wavelengths of the Global Ozone Monitoring by Occultation of Stars (GOMOS) operated on board the Envisat satellite during 2002-2012 (http://envisat.esa.int/instruments/gomos; Bertaux et al., 2010). Before the description of the measurements and inversion algorithm, we would like to define precisely what atmospheric parameter is retrieved (or "measured").

## 5   1.1   What is a high-resolution temperature profile?

In the case of the HRTP (high-resolution temperature profile), the underlying atmospheric parameter that we aim to characterize is the temperature field. However, the temperature is a four-dimensional scalar field, which is defined for each time moment over a three-dimensional (3D) space. Due to very active dynamical processes in the Earth's atmosphere, this 3D field can contain significant fluctuations down to the viscous scale, which is typically smaller (and sometimes much

smaller) than one meter within the stratosphere and the troposphere.

During occultations, the velocity of the sounding ray within the atmosphere is much larger than the velocity of any atmospheric motion (for GOMOS/Envisat, it is more than 3000 m/s), therefore the "frozen-field" approximation during measurement time can safely be considered (Tatarskii, 1971). Regarding the spatial variation, the trace of the line of sight within the atmosphere during an occultation defines a 2D surface, which differs only slightly from a plane because of

refractive effects. The only temperature fluctuations able to affect the GOMOS measurements lie in this surface. Furthermore, since the signal recorded by a detector is intrinsically one-dimensional, the retrieved parameter (temperature or density) is also one-dimensional.

Due to stable stratification of the stratosphere, most of the variations of meteorological parameters, such as the temperature, occur along the vertical direction. The field of temperature within the atmosphere is strongly anisotropic and the direction of

its gradient is close to the vertical. Consequently, a measurement of temperature variations along the vertical direction, known as a "temperature profile", describes most of the field variation within the considered region. However, the validity of the above statements is strongly dependent on the considered scale. Large-scale temperature fluctuations are strongly stratified (anisotropic), they contain the largest fraction of potential energy (or equivalently temperature variance). Most of the energetic dynamical processes, including meteorological flow and gravity waves correspond to this anisotropic part. The

characteristic ratio between dominant horizontal and vertical scales is typically equal to the ratio of maximal and minimal intrinsic frequencies of the gravity waves field. In the stratosphere, the ratio of buoyancy (Brunt-Väisälä) frequency $N$ to the Coriolis parameter $f$, $N/f$, is typically larger than 100 (Fritts and Alexander, 2003).

Small-scale fluctuations, mostly turbulence and, more generally, instable and dissipative processes, are much more isotropic (and so are the temperature gradients associated with such small-scale processes). For this kind of fluctuations, the concept

of a (vertical) profile is essentially meaningless. The transition between strongly anisotropic and roughly isotropic fluctuations occurs within the scale range separating the domains of waves and turbulence. For the stratosphere, it covers roughly a decade between 10 and 100 meters (of vertical scale) (Gurvich and Kan, 2003; Nastrom et al., 1997).



These general considerations about the structure of the atmospheric temperature field indicate that a one-dimensional "vertical profile" is only meaningful (for remote sensing measurements) for vertical scales larger than 30-100 m. The detailed characteristics of the measurement process must also be considered in order to grasp the real meaning of the retrieved profile and to understand better its relationship with the atmospheric temperature field. In case of GOMOS, the

concept of vertical profiles is adequate, as high-resolution temperature profiles, which will be discussed in our paper, have the vertical resolution of ~200 m.

## 1.2 Bi-chromatic scintillation measurements by GOMOS and previous works on HRTP

For retrievals of high-resolution temperature profiles, we use bi-chromatic scintillation measurements by the GOMOS fast photometers, which record the stellar flux with the sampling frequency of 1 kHz at blue (475-525 nm) and red (650-700

nm) wavelengths synchronously, as a star sets behind the Earth limb.

Two fast GOMOS photometers on board Envisat recorded the intensity fluctuations induced on the star's light by the refractivity fluctuations encountered within the atmosphere, at two wavelengths. A short description of the inversion algorithm for retrievals of high-resolution temperature profiles from bi-chromatic scintillation can be found in (Dalaudier et al., 2006; Sofieva et al., 2009c); it is presented also below in our paper.

The main advantage of HRTP is its vertical resolution, which is ~200-250 m. Such resolution allows probing gravity wave (GW) spectra. The validation of the small-scale structure of HRTP is therefore an important issue before using the data in GW research. The validation of the small-scale structure is a challenging task, because temperature fluctuations are rapidly varying due to gravity wave activity. Sofieva et al. (2008, 2009c) proposed to use spectral analysis for validation of small-scale structure in temperature profiles, as this approach allows using measurements separated by several hundreds of

kilometers and by several hours. The previous validation has been performed on HRTP processed by the FMI scientific processor (analogous to the ESA IPF v6 algorithm) using collocated radiosonde data. It has shown that the small-scale fluctuations in HRTP have similar rms as in collocated radiosonde profiles, for vertical (in orbital plane) occultations of bright stars (Sofieva et al., 2009c). In case of oblique occultations or dim stars, the HRTP fluctuations are of larger amplitude than those of in-situ measurements. An analogous study with the method (Sofieva et al., 2009c) but applied to much larger

datasets of GOMOS HRTP IPF v6 and collocated radiosonde profiles has shown that fluctuations in HRTP v6 are nearly always larger than in the collocated radiosonde data (for both vertical and oblique occultations). Therefore, IPF v6 HRTP data were not recommended for gravity wave analyses. This will be illustrated in Section 4 of this paper.

In this paper, we introduce an improved version of the GOMOS HRTP algorithm and present the whole GOMOS HRTP dataset processed using the FMI's Scientific Processor, HRTP FSP v1.

## 1.3 The paper structure

The basic principle of HRTP retrievals is described in Sect. 2. Section 3 is dedicated to a detailed description of the retrieval algorithm. Examples of retrieved HRTP profiles, their characterization and validation of the small-scale fluctuations are





shown in Sect.4. Illustrations of using HRTP for gravity wave analyses is presented in Section 5. The information about the GOMOS HRTP dataset and data access is presented in Section 6. Summary and discussion conclude the paper (Sect. 7)

## 2    Basic principle of HRTP retrieval

As discussed in (Kyrölä et al., 2010; Sofieva et al., 2009b), the light intensity transmitted through the atmosphere is affected

not only by absorption and scattering, but also by refraction and diffraction. The scintillations (or large intensity fluctuations) observed at the satellite level are the result of the interaction of stellar light and atmospheric air density irregularities, which are generated mainly by internal gravity waves and turbulence.

The retrieval of HRTP is based on the chromatic refraction in the atmosphere (Dalaudier et al., 2006). The refraction angle $\alpha$ depends on wavelength, due to the optical dispersion of air. For two rays of different color having the same impact parameter

$p$ (Figure 1), the blue ray bends more than the red one (Figure 1, $\alpha_B > \alpha_R$) and will consequently be observed later by GOMOS (only star settings are used in GOMOS observations). The scintillation spikes produced by atmospheric density fluctuations are observed by both photometers with a time delay $t_B - t_R$ (Figure 1), which is proportional to the refraction angle difference $\Delta\alpha = \alpha_B - \alpha_R$. The idea of such measurements of refractive angle has been first proposed by Gurvich and Sokolovskiy (1991, 1992).

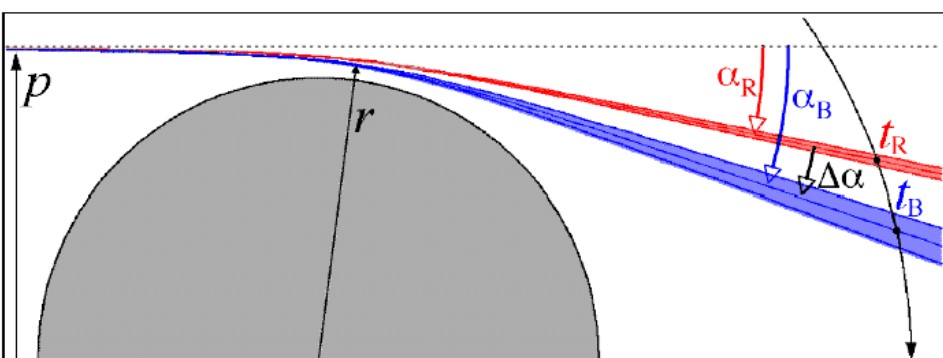

**Figure 1 A scheme of chromatic refraction and the principle of refraction angle measurement by GOMOS. Both the refraction angles and the effect of dispersion are strongly exaggerated. However, the width of each beam relative to the angle difference $\Delta\alpha$ is realistic. The impact parameter $p$ is the geometric distance of the ray from the Earth's center.  The vertical separation of blue and**
**red rays at  ray perigee 30 km is ~10 m** (Dalaudier et al., 2001)**.**

Using the accurate knowledge of the direction to the star  and of the ENVISAT orbit, it is possible to convert the measured time delay into an angle difference $\Delta\alpha$, and then into the refraction angle $\alpha$ at the reference wavelength (for GOMOS, the reference wavelength is 500 nm, i.e. $\alpha=\alpha_B$). The conversion factor is equal to $\alpha_B/\Delta\alpha = \nu_B^0/(\nu_B^0 - \nu_R^0) \approx 94$, where  $\nu_0(\lambda) = n_0(\lambda) - 1$ is the standard refractivity ($n_0$ is the refractive index) given for dry air at standard pressure and temperature in





(Edlen, 1966), retractivities $v_B^0$ and $v_R^0$ correspond to central wavelengths of GOMOS photometers. After that, the method is similar to that used in radio occultation (e.g., Kursinski et al., 1997). Assuming local spherical symmetry of the atmosphere, the refractive index profile $n(p)$ can be retrieved from the refraction angle profile $\alpha(p)$ using the Abel transform (Kursinski et al., 1997; Tatarskiy, 1968):

$$\log(n(p)) = \frac{1}{\pi} \int_p^\infty \frac{\alpha(q)\, dq}{\sqrt{q^2 - p^2}} \qquad (1)$$

The tangent (or minimal) radius $r$ is related to the impact parameter $p$ through the refractive index $n$ :

$p = r\, n = r\, n(p(r))$ .

The refractivity profile $v(r) = n - 1$ can be easily converted into a density profile through $\rho(r) = v(r)\rho_0 / v_0$ using the conversion factor for dry standard air (15°C, 101325 Pa). The corresponding pressure profile is reconstructed by integrating the hydrostatic equation, as it is done in radio-occultation and lidar measurements (e.g., Hauchecorne and Chanin, 1980; Kursinski et al., 1997). Finally, the temperature profile $T(r)$ is obtained from the state equation of a perfect gas.

## 3    From simplified theory to real experiment: HRTP processing algorithm

The main steps of processing the red and blue photometer signals to high-resolution temperature profiles can be outlined as follows:

- Estimation of the chromatic time delay as the position of the maximum of the cross-correlation function of blue and red photometer signals after smoothing the red one. Since refractive angle is proportional to chromatic time delay, the profile of the refractive angle is obtained

- Determination of the refractivity profile from the refractive angle profile via the Abel integral inversion. For upper limit initialization, an atmospheric model is used.

- The density profile is obtained from the refractivity profile using the Edlen's formula.

- From the density profile, the pressure profile is calculated using the hydrostatic equation.

- Finally, the temperature profile is determined from these data using the state equation of a perfect gas.

This basic algorithm is given for a highly simplified situation. Below we present the detailed description of the most important inversion steps and discuss various effects, which have occurred in the GOMOS experiment.





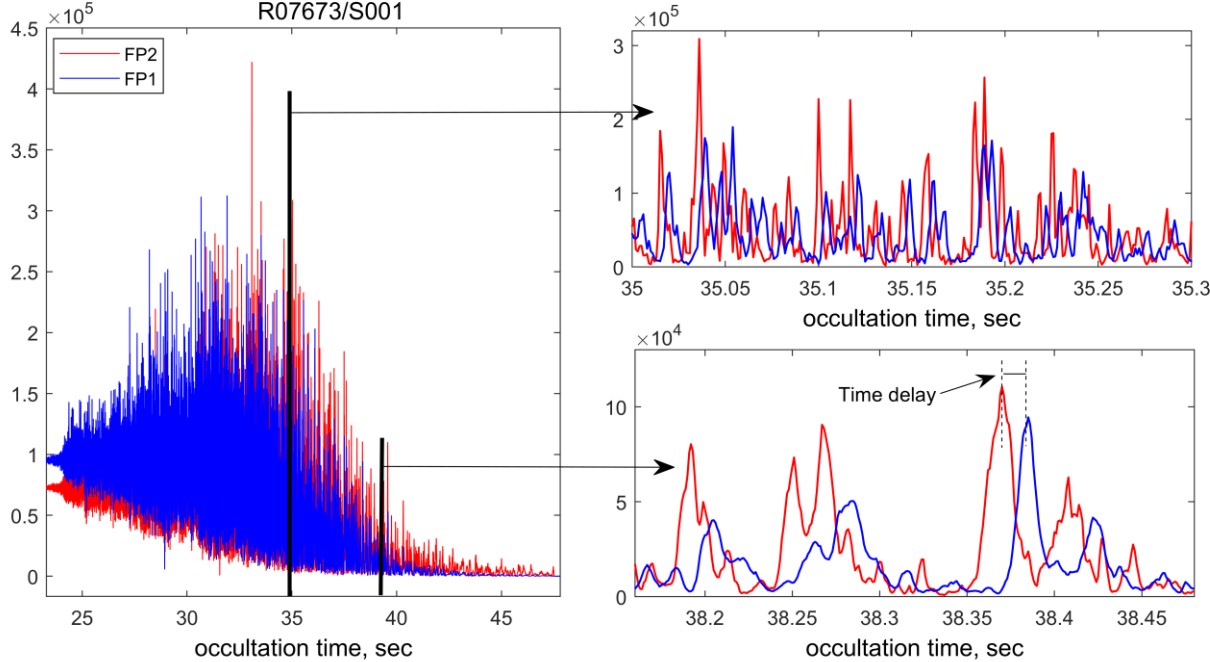

**Figure 2. Signals of red (FP2) and blue (FP1) GOMOS photometers.**

### 3.1 From photometer signals to the profile of time delay

5 The new HRTP processing starts at the altitude 32 km, where time delay is larger than 1 ms and the scintillation are of large amplitude. In the previous retrievals, the upper altitude, where the HRTP processing starts, depends on strength of scintillation and value of a priori time delay $\tau_a$ (estimated using ECMWF data). Since it is impossible to retrieve accurately time delay (and temperature profile) in the range where time delay is smaller than the sampling rate of the photometers, the upper limit is set to 32 km in the new retrievals.

10 The bandwidth of GOMOS photometers is nearly the same in wavelength, but the refractivity bandwidths of photometer optical filters are significantly different (Figure 3).

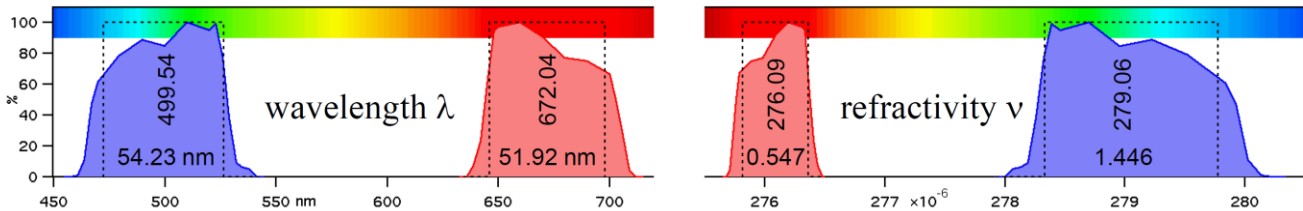

**Figure 3 Left: GOMOS optical filter response as a function of wavelength. Right: refractivity bandwidth of the GOMOS optical filters.**





As a result, scintillation features are more smoothed in the blue signal than in the red one (Figs. 1, 2). In order to make the chromatic smoothing similar, the signal of red photometer is convolved with a Gaussian window $G(t)$. The width of the smoothing window $W_G$ is defined as a differential width of the blue and red signals for a Dirac perturbation in the refractive angle, $W_G^2 = W_{blue}^2 - W_{red}^2$. It increases proportionally to the refractive angle as the line of sight deepens into the atmosphere and can be approximated as:

$$W_{\mathrm{G}} = \alpha_B L \frac{\sqrt{\Delta v_{0B}^2 - \Delta v_{0R}^2}}{v_{0B}} \frac{1}{\mathrm{d}h_{\mathrm{d}}/\mathrm{d}t}, \tag{2}$$

where $\Delta v_{0B}$ is the variation of the standard refractivity corresponding to the spectral width $\Delta \lambda_B$ of the blue photometer, $\Delta v_{0R}$ is defined analogously for the red photometer, $\mathrm{d}h_d/\mathrm{d}t$ is the vertical velocity of the line of sight and $L$ is the distance from tangent point to the satellite.

It is evident that the determination of time delay (as a function of time) by visual recognition of characteristic scintillation structures is not feasible. Therefore, the time delay is estimated via calculation of the cross-correlation function (CCF), followed by a determination of the position of its maximum. For computation of the CCF, photometer signals are cut into ~50 % overlapping sections. The length of the sections should be chosen in order to contain a "sufficient" number of structures (scintillation spikes) while preserving the best available resolution for the angle profile. We found that the optimal length of segments $\tau_{window}$ corresponds to a vertical displacement of the line of sight within the atmosphere varying from 250 m at 32 km to 500 m at 5 km. The smoothed red signal is pre-shifted by the (smooth) a priori time delay $\tau_a$ rounded to the nearest millisecond (which will be hereafter referred to as a pre-shifted time delay) in order to best align with the blue signal. For computation of $\tau_a$, we use ECMWF density data at the occultation location:

$$\tau_a(t) = \alpha \left( \lambda_B \right) L \frac{v_{0B} - v_{0R}}{v_{0B}} \frac{1}{\mathrm{d}h_d/\mathrm{d}t} \tag{3}$$

The CCF is calculated with 1 ms resolution and the position of its maximum is searched around zero delay, in the range $\pm(0.1\tau_{window} + 3)$ ms. The maximum point and its two neighbors are then interpolated using a parabola, which is the first approximation of the correlation function in the vicinity of its maximum. The examples of cross-correlation functions and their fits are shown in Figure 4 (right) for two GOMOS occultations. The position of the maximum of the parabola is then added to the pre-shift in order to provide the time delay estimation for the corresponding scintillation sample.

Uncertainty of time delay determination depends on the shape of the cross-correlation function and on how accurate it is. The error in the determination of the position of the cross-correlation function maximum can be estimated as

$$\sigma_\tau \approx \frac{\sqrt{2}\sigma_C}{\Delta t \, C''(\tau_0)} \tag{4}$$




where $C''$ is the second derivative of the cross-correlation function of photometer signals at the point of its maximum, $\sigma_C$ is the error (standard deviation) of the CCF, and $\Delta t$ is the discretization step (1 ms). The derivation of the formula (4) uses Taylor expansion of the condition for the CCF maximum is $C'(\tau) = 0$ at the vicinity of the maximum $\tau_0$ and the subsequent Gaussian error propagation. As follows from Eq. (4), uncertainty in determination of CCF maximum is

5 proportional to uncertainty of CCF values and depends on the shape of CCF: the broader the CCF, the larger error. In assumption of a Gaussian distribution of intensity fluctuations, the cross-correlation coefficient $C$ has an asymptotically Gaussian distribution with the standard deviation

$$\sigma_C \approx \frac{1 - C^2}{\sqrt{n}} \qquad (5)$$

where $n$ is the size of samples participating in calculation of the cross-correlation coefficient. This approximation is valid for

large samples. Finally, the error of time delay estimated can be written as

$$\sigma_\tau \approx \frac{\sqrt{2}(1 - C_{max}^2)}{C'' \Delta t \sqrt{n}} \qquad (6)$$

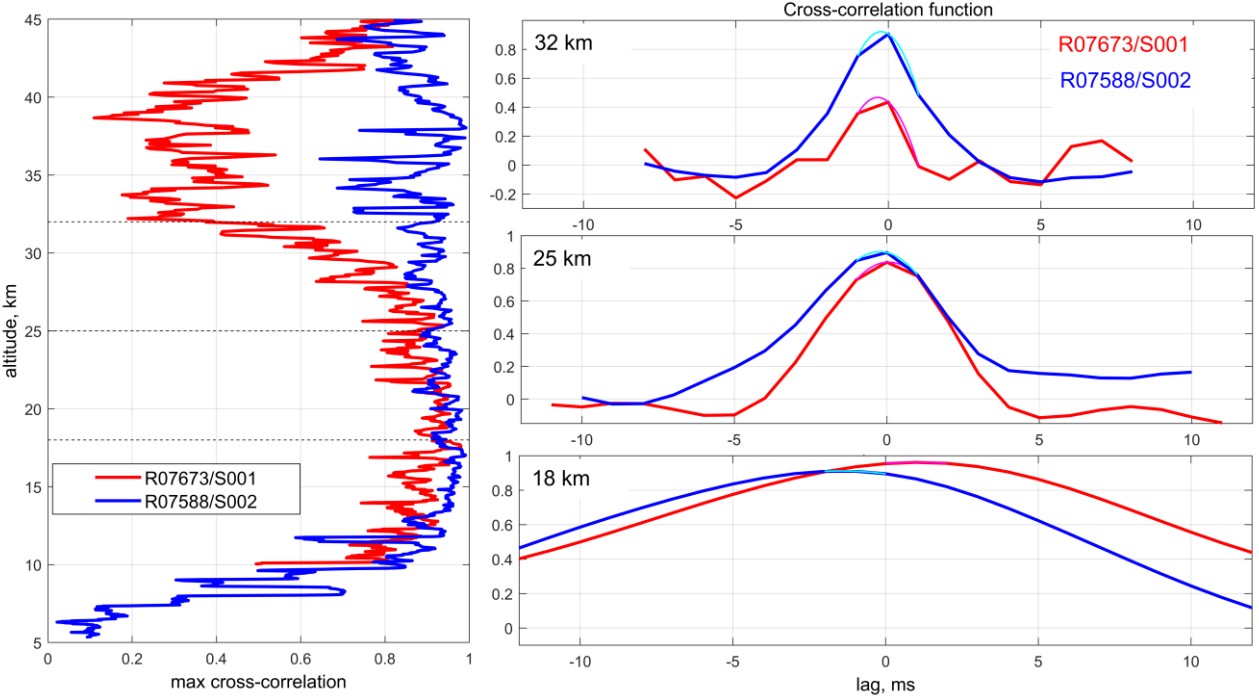

**Figure 4 Left: maximum of cross-correlation function for two GOMOS occultations: R07673/S001 (64°S 68°W, β=23°, 19-Aug-**

15 **2003 04:09:23), red line, and R07588/S002 (35°S 135°W β=-2.6° 13-Aug-2003 07:28:35), blue line. Right: examples of cross-correlation functions at selected altitudes and their fit with parabola.**





In (6), $C_{max}$ is the maximum of the cross-correlation function, or the cross-correlation coefficient; $C''$ can be calculated from parameters of the parabolic fit. Due to assumptions made in deriving Eq.(6), it is clear that this formula gives only an approximate estimate for the error of time delay reconstruction.

The uncertainty of time delay increases at lower altitudes (Figure 5, panel C, red line) due to broadening of cross-correlation function (Figure 4, right). At some levels, correlation between photometer signals can be low due to presence of turbulence, which leads to very uncertain time delay.

### 3.2 Regularization of time delay estimation

#### 3.2.1 Motivation: influence of isotropic turbulence

The atmosphere contains small-scale turbulence producing nearly isotropic fluctuations of density. These fluctuations also produce scintillation during stellar occultation. The contribution of turbulence to the observed scintillations can be significant and sometimes even dominant (Sofieva et al., 2007a, 2007b). The cross-correlation of bi-chromatic scintillations caused by isotropic turbulence is significant only when the chromatic separation distance of the ray trajectories does not exceed the Fresnel scale, which is ~1 m for GOMOS (for illustration and more details, see Fig. 4 and the corresponding text in (Sofieva et al., 2009b)). The chromatic de-correlation for nearly vertical (in orbital plane) occultations is always small, while it can be significant in case of oblique (off orbital plane) occultations (Kan, 2004). On the other hand, the smoothing induced by the finite optical band of the photometers will selectively damp the fluctuations associated with turbulence in the vertical direction because of their smaller size. As a result, the cross-correlation between the two photometers has a minimum at some altitude depending on the obliquity of the occultation (hereafter we define the obliquity angle β as is the angle between the direction of the apparent motion of the observed star and the local vertical at the ray perigee point, (Gurvich and Brekhovskikh, 2001; Sofieva et al., 2007b)). This is illustrated in Figure 4 (left) for the oblique occultation R07673/S001 with the obliquity angle β=23°. Strong turbulence is observed at upper altitudes, resulting in the drop of cross-correlation at 30-45 km. A more quantitative consideration of this effect is given in (Kan, 2004) and (Gurvich et al., 2005). In some situations, the correlation between photometers signals is too low for an accurate determination of the time delay (Figure 4 left). These situations are handled through regularization applied to the time delay profile, which is described below.

#### 3.2.2 Regularization algorithm

In the case of low correlation between the recordings of the fast photometers, the time delay determination as the position of the cross-correlation function maximum gives poor results, and it can induce unphysical fluctuations in the time-delay profile.



In the V6 algorithm, the data points corresponding to low correlation between photometer signals (with cross-correlation coefficient CCC<0.7) are replaced by the weighted mean of "measured" (obtained from cross-correlation) $\tau_{meas}$ and a priori (computed external data source, e.g. ECMWF analysis data) $\tau_a$ time delays:

$$\hat{\tau} = \frac{\tau_a / \sigma_a^2 + \tau_{meas} / \sigma_\tau^2}{1 / \sigma_a^2 + 1 / \sigma_\tau^2}, \qquad (7)$$

5  The weights used in (7) are inversely proportional to the uncertainties of time delay $\sigma_\tau^2$ (defined by Eq.(6)) and the a priori profile $\sigma_a^2$. Hereafter, we will refer to this regularization as to the optimal filtration method. In the HRTP IPF v6 algorithm, the uncertainty of the a priori time delay is computed assuming that a priori air density has an uncertainty of 2.5% below 25 km, 5 % at 35-50 km with the linear transition between these two altitude regions. The effects of optimal filtration on time delay, its uncertainty, and the resulting temperature profile in illustrated in Figure 5 by blue lines.  As observed in Figure 5,

10  such filtration handles exceptional values, but the resulting temperature profile has enhanced amplitude of temperature fluctuations at altitudes below 17 km compared to collocated sonde data. Validation of HRTP profiles, which have been processed with optimal filtration, against collocated radiosonde data (Sofieva et al., 2009c) has shown that the amplitude of temperature fluctuations in HRTP is realistic for vertical occultations of bright stars (not affected by turbulence), but often excessive in oblique occultations.

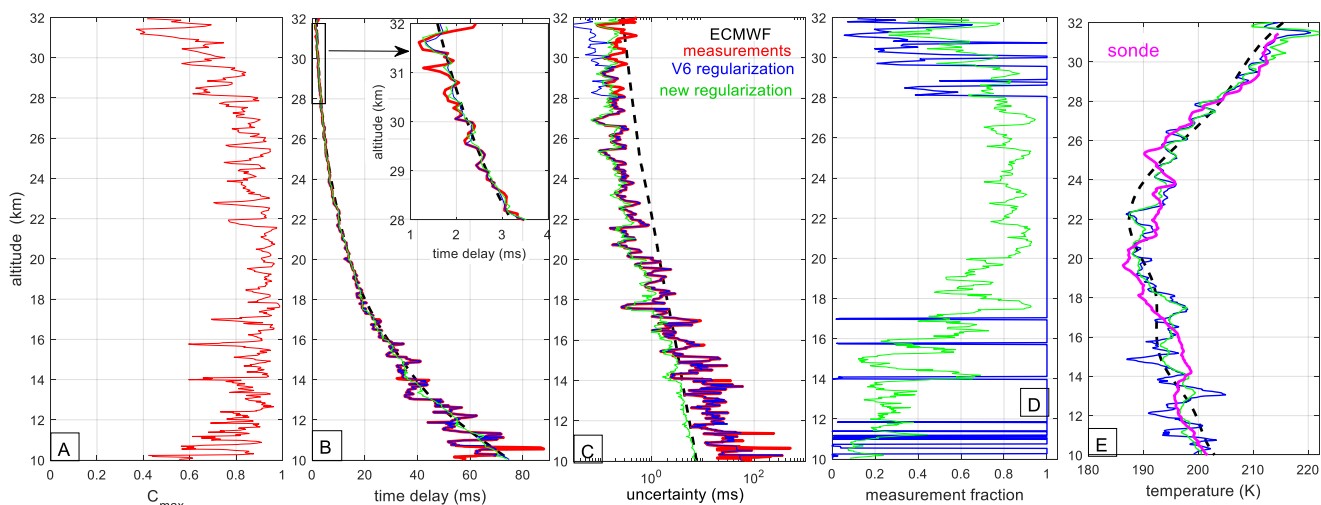

**Figure 5. Illustration of time delay regularization and retrievals based on occultation R07673/S001; red: pure measurements, blue: V6 regularization, green: new (Bayesian) regularization, black: a priori (ECMWF). A: profile of cross-correlation coefficient C$_{max}$. B: profiles of time delay, C: uncertainty of time delay; D: measurement fraction, E: retrieved temperature profiles and collocated sonde measurements at Marambio (magenta line)**





In new algorithm, we use the statistical optimization (Bayesian approach). It acts as a linear operator applied to the differences between measured and a priori time delays:

$$\tau_{reg} = \tau_a + \mathbf{C}_a(\mathbf{C}_a + \mathbf{C}_{meas})^{-1}(\tau_{meas} - \tau_a) \tag{8}$$

where $\mathbf{C}_a$ is the covariance matrix of a priori time delay uncertainties and $\mathbf{C}_{meas}$ is the covariance matrix of measured time delay uncertainties. This formulation corresponds to the Bayesian estimator (maximum a posteriori method) provided measurement errors and a priori uncertainties have Gaussian distribution. If both matrices $\mathbf{C}_a$ and $\mathbf{C}_{meas}$ are chosen to be diagonal, the Bayesian approach coincides with the optimal filtration (7). The diagonal elements of $\mathbf{C}_a$ and $\mathbf{C}_{meas}$ are "a priori" and "measured" uncertainties (variance of the corresponding errors), while off-diagonal elements characterize the correlation length scale of measurements and a priori profiles. In the new algorithm, the correlation length for measurements is equal to the window used for computation of cross-correlation function, while the correlation length of a priori profile is set as twice larger. The diagonal elements of $\mathbf{C}_a$ and $\mathbf{C}_{meas}$ are $\sigma_a^2$ and $\sigma_\tau^2$, as in the optimal filtration. The covariance matrix of the regularized time delay is estimated as

$$\mathbf{C}_{reg} = \left( \mathbf{C}_a^{-1} + \mathbf{C}_{meas}^{-1} \right)^{-1}. \tag{9}$$

With such approach, the uncertain values of time delay are replaced not at selected altitudes like in V6 algorithm, but according to length of photometer record used in determination of time delay. The resulting time delay, its uncertainty, and final temperature profile is illustrated in Figure 5, green lines. The resulting profile of time delay follows measurements in the stratosphere, while at lower altitudes, where the uncertainty of measurements is large, it follows a priori profile. The measurement fraction shown in Figure 5 (panel D) is defined as

$$F = \frac{\mathbf{C}_a(\mathbf{C}_a + \mathbf{C}_{meas})^{-1}\tau_{meas}}{\tau_{reg}} \tag{10}$$

In the stratosphere, it is close to 1, while it decreases at lower altitudes.

### 3.3 From time delay to refractive angle

Refractive angle is proportional to time delay between photometer signals. The proportionality coefficient depends on difference in refractivity corresponding to the central wavelengths of photometers, distance from ray perigee point to the satellite, and the satellite velocity (Eq. (3)). However, this simple relation is complicated by the fact that the stellar spectrum is modified by absorption and scattering during an occultation. As a result, the effective wavelength of the photometer signal varies with altitude. Knowing the stellar spectrum $I(\lambda, t)$ measured through the atmosphere and the transmission functions $f_{filter}(\lambda)$ of the photometer optical filter, it is possible to determine the effective wavelength:



$$\lambda_{\text{eff}}(t) = \frac{\int_{\lambda_{\min}}^{\lambda_{\max}} \lambda I(\lambda,t) f_{\text{filter}}(\lambda) d\lambda}{\int_{\lambda_{\min}}^{\lambda_{\max}} I(\lambda,t) f_{\text{filter}}(\lambda) d\lambda} \quad , \tag{11}$$

where $\lambda_{\min}$ and $\lambda_{\max}$ correspond to the wavelength range of the optical filter.

Then the refractive angle $\alpha$ at the reference wavelength, which will be used in the further processing, can be computed as:

$$\alpha = \frac{\tau}{L} dh_{\text{d}} / dt \frac{v_0(\lambda_{\text{ref}})}{v_0(\lambda_{\text{eff,blue}}) - v_0(\lambda_{\text{eff,red}})} \tag{12}$$

5 Since the refractive angle is proportional to time delay, its uncertainty can be easily obtained by multiplication of time delay uncertainty by the corresponding factor.

## 3.4    From refractive angle to refractivity profile

By purely geometrical considerations and because the refractive angle is small, the impact parameter $p$ for a given wavelength $\lambda$ is given by:

$$p(\lambda,t) = h_{\text{d}}(t) + \alpha(\lambda,t) L(t) \tag{13}$$

Applying the inverse Abel transform (Eq. 1), we can obtain the profile of refractive index $n(p)$. The application of the Abel transform assumes local spherical symmetry of the atmosphere. This assumption is also used in retrievals from radio-occultation measurements. The error due to horizontal gradients of the refractive index at right angles to the direction of light propagation has been estimated in (Healy, 2001; Sofieva et al., 2004); it is less than 1 % for altitudes. The integration of Eq.

15 (1) can be carried out numerically using any standard quadrature method. The weak singularity of the integrand at the lower limit does not cause problems for a numerical realization: the singularity can be estimated or the midpoint product integration method can be applied. The upper limit should be chosen high enough (~120 km), therefore the refractive angle profile calculated using ECMWF&MSIS data is used at altitudes above HRTP range in the processing. Application of Abel integration requires monotonous impact parameter. This requirement can be violated, because the impact parameter is

20 computed using measured (noisy) refractive angle. In the current implementation, the impact parameter is computed using the smoothed refractive angle and its monotonicity is checked.

Real geometric (tangent) altitudes can be determined as

$$h = \frac{p}{n(p)} - R \tag{14}$$

where $R$ is the local radius of curvature of the Earth surface.



The error of refractivity reconstruction can be estimated using the matrix of the discretized Abel transform. Due to the fact that the Abel integral acts as a linear operator connecting refractive angle and refractivity, the covariance matrix of refractivity uncertainty $\mathbf{C}_v$ can be estimated using the classical error propagation formula:

$$\mathbf{C}_v = \mathbf{A}\mathbf{C}_\alpha\mathbf{A}^{\mathrm{T}} \tag{15}$$

where $\mathbf{A}$ is the matrix of the discretized Abel transform (see e.g. Sofieva and Kyrölä, 2004) and $\mathbf{C}_\alpha$ is the covariance matrix of refractive angle uncertainties.

### 3.5    From refractivity to density, pressure and temperature

The density profile can be obtained from the refractivity profile using Edlen's formula. By using the hydrostatic equation we can calculate the pressure $P$ at the altitude $h$ as

$$P(h) = \int_h^\infty g(x)\rho(x)dx, \tag{16}$$

where $g(x)$ is the acceleration of gravity. The high altitude initialization of pressure is obtained from an external model.

Finally, temperature can be determined from the equation of state of a perfect gas

$$T = \frac{\mathrm{M}}{\mathrm{R}}\frac{P}{\rho} \;, \tag{17}$$

where R=8.3144 J/mol/K is the universal gas constant and M is the molar mass of dry air.

The covariance matrix of air density errors $\mathbf{C}_\rho$ is proportional to the covariance matrix of refractivity errors $\mathbf{C}_v$ (relative errors are equal).

Two main terms contributing to the error in temperature are the error in local density (small scale structures in density and temperature are anti-correlated and of equal relative amplitudes) and the error in pressure at the top of the high resolution profile $P_{\mathrm{top}}$ (error of upper limit initialization). They are added quadratically, thus giving the uncertainty of HRTP:

$$\left(\frac{\Delta T}{T}\right)^2 = \left(\frac{\Delta\rho}{\rho}\right)^2 + \left(\frac{\Delta P_{\mathrm{top}}}{P_{\mathrm{top}}}\frac{P_{\mathrm{top}}}{P}\right)^2. \tag{18}$$

In HRTP retrievals, the vertical resolution is defined mainly by the length of scintillation records that are used for calculation of cross-correlation function, which is ~ 250 m. Due to using overlapping samples, the actual vertical resolution is somewhat smaller. The regularization on time delay, as well as other inversion steps from time delay to temperature profile, can slightly degrade the vertical resolution in case of oblique occultations, thus overall vertical resolution of HRTP is expected to be close to 250 m.



## 4    Retrieved HRTP profiles, their characterization and validation

Examples of retrieved GOMOS high-resolution temperature profiles are shown in Figures 6 and 7 by red lines with 1σ uncertainties (shaded area). In Fig.6, HRTP profiles are for bright stars in vertical occultations (the best data quality), while in Fig. 7 other occultations are illustrated (oblique or of not bright stars). These temperature profiles are collocated with high-resolution radiosonde data from the SPARC data center (http://www.sparc.sunysb.edu/html/hres.html). The collocated temperature profiles are shown by blue lines in the left panels of Figs. 6 and 7, and the information about the spatio-temporal difference is provided in the figure. We would like to note that the fine structure in the HRTP and radiosonde profiles are not expected to coincide, because of the evolution of the gravity wave field in the space-time window. For similarity of temperature profiles, including their small-scale fluctuation, the horizontal separation should be ideally less than 20 km and the time difference should not exceed 2-3 h, as discussed in Sofieva et al. (2008, 2009c). The time separation results in additional spatial separation in the atmosphere caused by advection of air masses. The relatively long measurement time of temperature profiles by radiosondes during balloon flights (it takes ∼1 hour for balloon to raise to 10–30 km) has a similar effect (the quantitative estimates of these effects can be found in Sofieva et al. (2009c). In the left panels of Figures 6 and 7, the HRTP and collocated radiosonde profiles are similar, but not fully coinciding, as expected.  The temperature profiles for vertical occultation of bright stars are of similar quality as in other occultations, as follows from comparison of Figures 6 and 7.

Despite differences in small-scale temperature fluctuations, we can expect similar spectral properties of the temperature field at locations not far from each other (e.g., less than 500 km) during some time period (a few hours), as shown in (Sofieva et al., 2008). The power spectra density of relative temperature fluctuations in HRTP and collocated radiosonde profiles are shown in the right panels of Figures 6 and 7. For the spectral analysis, the collocated profiles were interpolated to an equidistant altitude grid with a 30 m resolution in the altitude range 18-30 km. Hanning filtering with a 3 km cut-off scale was used to obtain the smooth component. One can notice a good agreement of wavenumber spectra of HRTP and radiosonde temperature fluctuations, for all the considered GOMOS occultations. In contrast to the previous HRTP validation results, the spectra of temperature fluctuations are similar also in case of oblique occultation or not bright stars.

The HRTP wavenumber spectra in Figures 6 and 7 have visible cut-offs corresponding to scales ~150-250 m. This is the experimental conformation of the vertical resolution of HRTP ~200 m. This agrees with the theoretical estimates of the HRTP retrievals (it is defined mainly by the lengths of the scintillation records used for evaluation of time delay).





**Figure 6.** Left: examples of HRTP and collocated radiosonde profiles for some full-dark vertical occultations of very bright stars: Right: power spectral densities for these profiles, for the altitude range 18-30 km. Dashed black lines show the spectra corresponding to the saturated gravity waves model with the slope -3 (Smith et al., 1987). The information about GOMOS measurements, obliquity angles β, star magnitude mv, spatial Δd and temporal Δt separation, as well as the values of rms fluctuations are specified in the Figure.







**Figure 7.** As Figure 6, but for oblique occultations or/and not bright stars.




Typical estimated (in the retrieval algorithm) uncertainties of HRTP retrievals are shown in Figure 8, for occultations of different types. The HRTP temperature profiles are considered in the equatorial region (20°S-20°N, tropopause is ~18 km) in 2004.  As seen in Figure 8, the estimated HRTP uncertainty is 1-3 K in the stratosphere, at altitudes from ~ 2 km above the

5    tropopause to ~30 km. The best quality is achieved in vertical occultations. The uncertainty of the retrievals depends weakly on star brightness, it is noticeably larger only for occultations of dim stars with visual magnitude mv>2.5.

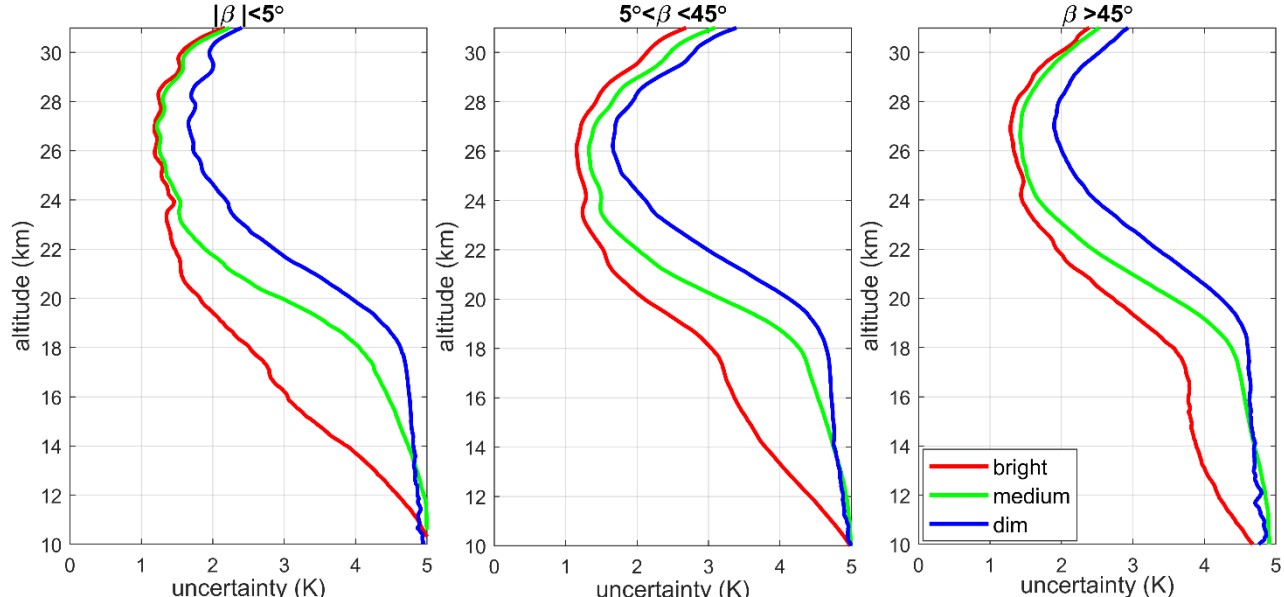

**Figure 8. Mean uncertainties of HRTP in the equatorial region 20°S-20°N in 2004, for different obliquityangles β specified in the panels, and for different stars: bright (mv<1), medium (1<mv<2.5) and dim (mv>2.5).**

    In this paper, we focus on the validation of small-scale fluctuations in the HRTP, as the HRTP vertical resolution allows probing gravity waves.  The validation of small-scale structure is a challenging task, because temperature fluctuations are rapidly varying due to gravity wave activity. Following Sofieva et al. (2008, 2009c), we use spectral analysis for validation of small-scale structure in temperature profiles, as this approach allows using measurements separated by several

15    hundreds of kilometers and by several hours. This study applies the method by Sofieva et al. (2009c) to much larger datasets of  HRTP and collocated radiosonde profiles.

    We have used the radiosonde data from the US high-resolution radiosonde archive available at http://www.sparc.sunysb.edu/html/hres.html (SPARC data center). The US high-resolution radiosonde archive contains data from 93 US operated stations from years 1998-2011. The stations are located across the mainland US, Alaska, Pacific islands

20    and the Caribbean. Most of the data is in 6-second  temporal resolution (the vertical resolution is ~30 m), but in recent years,





the stations have been upgraded to provide data in 1-second resolution. Only the 1-second data are used in the analysis presented in this paper, however the US 6-second data and other high-resolution radiosonde data sets obtained from NDACC, NILU, SHADOZ and Sodankylä radiosonde station are also analyzed and they show similar results. The collocated HRTP and radiosonde data are selected using 300 km and 4 h space-time window. With this criterion, 5023 collocated

profiles are found in the 6-second resolution data set and 1070 in the 1-second resolution data. Among the collocated data, there are occultations of different types.

Having in mind the results of the previous validation, we compared the rms of temperature fluctuations in radiosonde temperature profiles with those in HRTP, in the altitude range 18-30 km, and for different occultation types: vertical ($|\beta| < 5°$), of medium obliqueness ($5° < \beta < 45°$), and highly oblique ($\beta > 45°$), and for stars of different

brightness: bright (visual magnitude $m<1$), of medium brightness ($1<m<2.5$), and dim ($m>2.5$). We detected fluctuations about the smooth profile, which was computed from the original profiles using a Hanning filter with the 3 km cut-off scale. The rms of temperature fluctuations in collocated HRTP and radiosonde data, $\sigma_{hrtp}$ and $\sigma_{sonde}$, respectively, are presented as scatter plots (Figure 9). The colored markers in Figure 9 correspond to different versions of HRTP processing: IPF v6 (green), v6 algorithm implemented in FMI scientific processor (blue), and the new HRTP algorithm presented in this paper

(red). We found that rms of HRTP v6 fluctuations is overall larger than that in collocated radiosonde profiles for all occultation types, despite the vertical resolution is finer for radiosonde data (and thus the opposite behavior is expected). However, the v6 algorithm implemented in FMI HRTP scientific processor shows the behavior consistent with the previous analysis of (Sofieva et al., 2009c): the small-scale fluctuations in HRTP are realistic for vertical occultations of bright stars, and the HRTP fluctuations are of larger amplitude than in collocated radiosonde temperature profiles in case of oblique

occultations or dim stars. For the new HRTP retrieval algorithm, $\sigma_{hrtp}$ and $\sigma_{sonde}$ are similar for all occultation types. This demonstrates a clear improvement of new HRTP data.







**Figure 9. Standard deviations of temperature fluctuations in the altitude range 18-30 km for occultations of different types (see text for explanation). Processing version: IPF V6 (green), V6 algorithm implemented in the FMI scientific processor (blue) and new HRTP-FMI algorithm (red). The dashed black lines: $y = 1.2x$ and $y = (1/1.2)x$, solid black lines: $y=1.4\,x$ and $y=(1/1.4)x$.**

5    Several examples of wavenumber spectra of relative temperature fluctuations are shown in Figures 6 and 7, which show quite typical behavior: the spectra are similar for HRTP and collocated radiosonde profiles.



## 5 Illustrations of HRTP application: GW potential energy

In this section, we show illustrations of applications of HRTP for analyses of gravity waves. The spatio-temporal distributions are presented only for illustrations that new HRTP dataset provide valuable geophysical information, which is in agreement with analyses using other datasets.

The gravity wave potential energy per unit mass is defined as

$$E_P = \frac{1}{2}\frac{g^2}{N^2}\left\langle \left(\frac{\delta T}{T_s}\right)^2 \right\rangle,$$

(19)

where $\dfrac{\delta T}{T_s}$ are relative temperature fluctuations with respect to the smooth (background) profile $T_s$, and $g$ is acceleration of gravity.

In our analysis, smooth profiles $T_s$ are obtained by smoothing HRTP down to 4 km resolution. The Brunt-Väisälä frequency

$N^2$ is estimated using the smoothed HRTP profile. The GW potential energy has been evaluated for each temperature profile in the altitude range 20-30 km.

Figure 10 shows the distribution of GW energy in two seasons, winter and summer. These distributions are evaluated using individual $E_p$ values averaged in 10° latitude ×20° longitude bins, for the whole GOMOS dataset from 2002 to 2011. The distributions of $E_P$ shown in Figure 10 are in very good agreement with previous estimates of this parameter from

radiosonde, lidar and GPS radio-occultation measurements, both qualitatively and quantitatively (Allen and Vincent, 1995; de la Torre et al., 2006; Sofieva et al., 2009a; Tsuda et al., 1991, 2000). The overall distribution reproduces the known features: strong GW activity close to the edge of polar vortex, a peak near Antarctic Peninsula in local winter. Enhancements in equatorial regions are clearly observed, analogous to those found in global analyses of radio-occultation data (de la Torre et al., 2006; Tsuda et al., 2000).

Temporal evolution of GW potential energy, for different latitudes, is shown in Figure 11. This time series is very similar to that shown in (de la Torre et al., 2006, Figure 1) obtained from radio-occultation data. Enhancements at polar and mid-latitudes in winter are observed; they are larger in the Southern Hemisphere and follow the evolution of polar night jet (de la Torre et al., 2006; Sofieva et al., 2009a). The enhancements in the equatorial region is also observed, which seem to be annual but might be also modulated by quasi-biennial oscillations. Similar equatorial enhancements are observed by de la

Torre et al. (2006).

We would like to note that, despite similarities of the GW energy morphology presented in our paper with the previous studies, there are expectedly some differences and peculiar features, because also contributions of small-scale gravity waves (down to 250 m vertical scales) are present in HRTP profiles. Detailed analyses of gravity wave distributions using HRTP might be the subject of future works and publications.





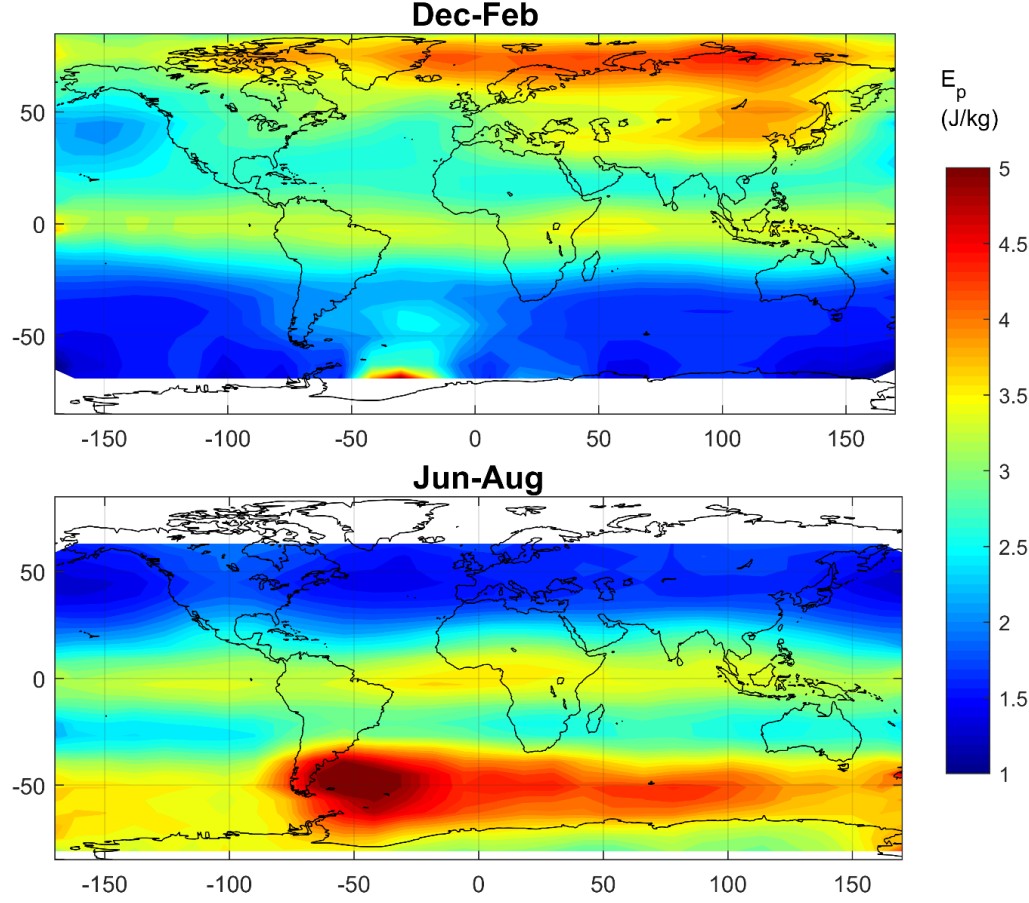

**Figure 10. GW potential energy in two seasons, in years 2002-2011.**

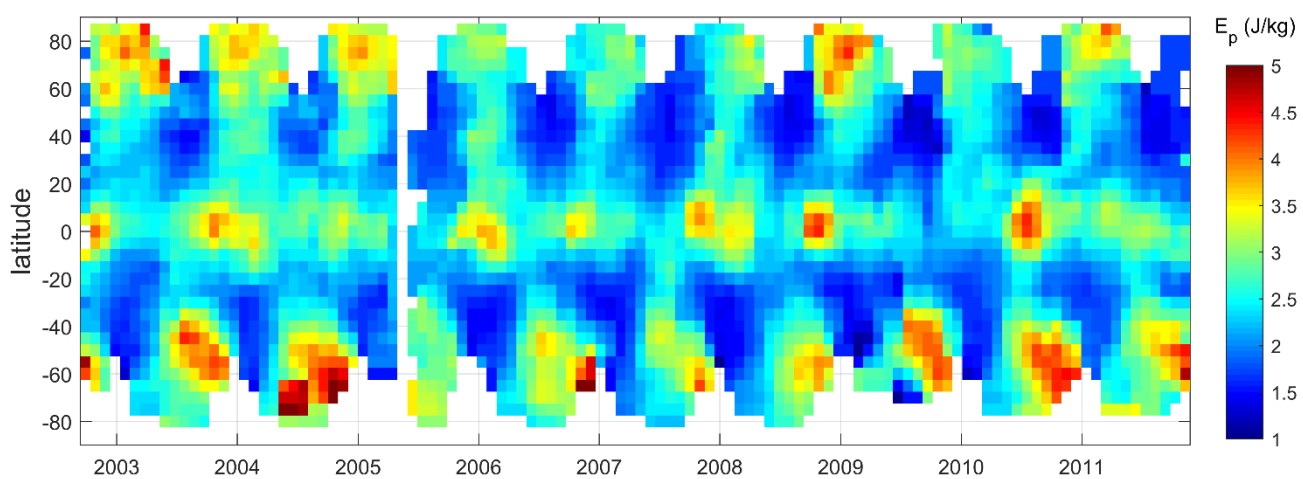

**Figure 11. Time series of GW potential energy (zonal average) estimated from GOMOS HRTP in years 2002-2011.**



## 6    Summary and discussion

We described the improved algorithm for high-resolution temperature and air density profiling using the bi-chromatic scintillation measurements. This method allows temperature profiling in the altitude range 10-35 km with the vertical resolution of ~ 250 m and accuracy in the stratosphere of 1-2 K. The retrieval algorithm is applied to the whole GOMOS/Envisat dataset, and the new GOMOS HRTP FSP v1 data set is now available. The best accuracy is achieved for vertical occultations bright stars. The uncertainty of HRTP retrievals depends weakly on stellar brightness.

The spectral analysis of HRTP and collocated radiosonde profiles has been applied for validation of small-scale fluctuations. It has been shown that the HRTP fluctuations are realistic (in terms of their 1D vertical spectra).

The main factors limiting accuracy of HRTP retrievals are due to instrumental properties in combination with the specifics of refraction in the Earth atmosphere. The upper limit of HRTP is defined mainly by the sampling frequency of the photometers: the detectable time delay should be larger than the photometer integration time, 1 ms; it is usually below 35-37 km. For faster photometers and larger wavelength separation, the upper limit can be higher. The presence of uncorrelated scintillations generated by locally isotropic turbulence reduces the useful information content in the photometer data. At lower altitudes, the influence of isotropic turbulence is low due to selective filtering by the photometer optical filters. However, lower signal-to-noise ratio at lower altitudes due to influence of absorption and broadening of scintillation peaks due to chromatic smoothing degrade accuracy of HRTP retrievals at altitudes below 15-17 km. Narrower optical filters would  allow slightly better retrievals an lower altitudes. The physical model for HRTP retrievals is adapted for vertical and moderately oblique occultations (for which $\tan(\beta)$ is smaller than anisotropy of air density irregularities). Such occultations constitute absolute majority of GOMOS measurements. The crossing of rays (strong scintillation) at low altitudes is not taken into account by the model. However, its influence is reduced due to the spatio-temporal averaging by the GOMOS optical filters (Kan et al., 2001), thus allowing acceptable temperature retrievals from GOMOS scintillation measurements also at altitudes below 25 km.

HRTP can be assimilated into atmospheric models, used in studies of stratospheric clouds and in analysis of internal gravity waves activity. As an illustration of application of HRTP for gravity wave research, GW potential energy has been evaluated using the GOMOS HRTP dataset. The obtained spatio-temporal distributions of GW potential energy are in good agreement with previous analyses using other datasets.

This paper is dedicated mainly to the retrievals, validation and geophysical assessment of small-scale fluctuations in the retrieved GOMOS high-resolution temperature profiles. However, HRTP can be smoothed down to lower resolution, and used in other analyses, including analyses of temperature trends, in combination of 10-years GOMOS HRTP data with other limb profile temperature measurements. This can be a subject of future research.



**Data availability**

The HRTP dataset is available from http://ikaweb.fmi.fi. The HRTP profiles presented in the dataset are interpolated to a common altitude grid from 10 to 32 km with 50 m spacing and stored in yearly netcdf-4 files. The README document provides the information about the parameters included in the data files.

**Acknowledgements**

The authors thank ESA for the GOMOS data and support through a dedicated project on improvement of HRTP data. The thank the whole GOMOS team for many years of fruitful collaboration. The work of V.F. Sofieva has been supported by the Academy of Finland, Centre of Excellence of Inverse Modelling and Imaging. The work of V. Kan has been supported by
the Russian Foundation for Basic Research, grant 16-05-00358. We thank the Academy of Finland, project TT-AVA, for supporting the collaboration between FMI and A.M. Obukhov Institute of Atmospheric Physics, Russia. The authors thank the FMI summer student Teemu Tiinanen for his contribution to HRTP analyses.

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
