# Peer review of "High-resolution temperature profiles retrieved from bi-chromatic stellar scintillation measurements by GOMOS/Envisat"

_Atmospheric Measurement Techniques, 2018_

## Referee Comment (RC1) · Anonymous Referee #1 · 30 Oct 2018

The manuscript presents the inversion algorithm for the processing of high resolution temperature profiles from GOMOS measurements. The algorithm exploits the time delay of scintillations observed from photometers at two different wavelengths to obtain vertical refractive index profiles, which are then transformed to temperature using hydrostatic balance. The retrieved profiles are compared to colocated radiosonde data and are shown to have improved spectral features relative to other GOMOS temperature products. The full GOMOS mission has been processed with the new algorithm and the dataset is in open access.

The manuscript is well organized and the logical progression of the retrieval is easy

to follow. I believe the manuscript is well suited for AMT and is suitable for publication following taking into account my comments below.

**General Comments**

Improvements could be made to better explain/quantify the effect of regularization on the time delay signal. I found it difficult to intuitively understand what the measurement fraction (Eq. 10) is saying in relation to other commonly used metrics of the effect of regularization. Since the time delay regularization approach is essential an optimal estimation style linear retrieval with $K = I$, would it instead be possible to calculate the equivalent averaging kernel and show rows/their sum? This would also give a sense of what effect the regularization has on the vertical resolution of the time delays.

A major point of the manuscript is the validation of small scale structure in the new HRTP FSP v1 data, and in particular its improvement over other processors. This is obviously a difficult task, but the validation efforts presented do show that the new data is more consistent with the radiosonde data than that of other processors. The part that I feel is missing is an explanation of why that improvement occurs. Differences in the regularization/upper limits are clearly outlined, however are these the only differences or are there others? What is the difference between the IPF and FMI v6 processing? The manuscript would be improved by a brief overview of how the presented retrieval method is different than the IPF v6 and FMI v6 processing and how these differences can explain the observed improvement. If the primary reason for the improvement is the different regularization scheme then I think the effect of regularization needs to be better quantified, perhaps with the averaging kernel suggestion above.

[Figure]

**Minor Comments**

p.1 l.17: "The HRTP profiles are retrieved with . . . high accuracy of ∼1-3 K"
As far as I can tell there is no discussion of accuracy anywhere in the discussion, this statement seems to apply to the HRTP precision. There are several other places in the abstract where accuracy should read precision.

p.2 l.16: "Furthermore, since the signal recorded by a detector is intrinsically one-dimensional, the retrieved parameter (temperature or density) is also one-dimensional." I think this statement is slightly misleading, it should be made clear that this is an approximation being made (the next paragraph does in fact justify this approximation).

p.2 l.26: "The ratio of buoyancy frequency $N$ to the Coriolis parameter $f$, $N/f$, is typically larger than 100."
The ratio of limb horizontal path length to vertical path length is also about 100, does this mean that large scale horizontal fluctuations are expected to have an effect on GOMOS measurements?

p.3 l.20: "... HRTP processed by the FMI scientific processor (analagous to the ESA IPF v6 algorithm) . . . "
This is is the only statement I could find about the relationship between the ESA and FMI v6, however from Fig. 9 there appears to be large differences between the two.

p.6 l.6: "In the previous retrievals, the upper altitude, where the HRTP processing starts, depends on the strength of scintillation and value of a priori time delay $\tau_a$"
An additional statement is needed here as to why this was changed to be fixed at 32

km. Presumably in the previous version of the retrieval altitudes where the time delay was expected to be less than the sampling rate of the photometers was not used. Was it a problem with the a priori data not being accurate? Or something else?

p.9 l.16: "These situations are handled through regularization applied to the time delay profile . . . "
I initially read this and was confused. The word regularization in standard limb retrievals is used because there is not enough information to retrieve the (essentially continuous) target quantity, but here the time delay can be found exactly. That being said, the technique applied is a standard regularization technique for limb retrievals and has a simple intrepretation as 'virtual measurements' added to the system. I am not sure if it would be better to use a different word (nothing immediately comes to mind), but it should be made clear that this is not regularization based upon the standard mathematical definition.

p.11 l.6: "If both matrices $C_a$ and $C_{meas}$ are chosen to be diagonal, the Bayesian approach coincides with optimal filtration"
This is only the case if what is referred to as "optimal filtration" does not include the CCC $< 0.7$ criteria that was used in v6. In fact the effect of removing this condition might be more important than going to full covariance matrices, is there an estimate of which of these two things is the dominant difference?

p.12 l.5: "Since the refractive angle is proportional to the time delay, its uncertainty can be easily obtained by multiplication of the time delay uncertainty by the corresponding factor."
I assume that the uncertainty in the spectra is negliglble compared to uncertainty in the time delay?

p.12 l.13: "The error due to horizontal gradients of the refractive index at right angles to the direction of light propagation has been estimated . . . it is less than 1% for altitudes"
Is this for all altitudes or the altitudes relevant for the retrieval?

p.12 l.18: "ECMWF & MSIS data is used at altitudes above HRTP range in the processing"
Is there an estimate for how much of an effect this has on the final retrieved temperature?

p.13 l.23: "The regularization on time delay, as well as other inversion steps from time delay to temperature profile, can slightly degrade the vertical resolution ..."
The effects here that can be quantified should be, in particular the effect of regularization.

p.14 l.3: ". . . bright stars in vertical occultations . . . "
Vertical here means in orbital plane?

p.14 l.5: ". . . from the SPARC data center (http://www.sparc.sunysb.edu/html/hres.html)"
Link did not work for me when I tried it, but it might have just been temporarily down.

p.14 l.26: "The HRTP wavenumber spectra in Figures 6 and 7 have visible cut-offs corresponding to scales ∼150-250 m"
It is hard to see this from the figure, maybe something could be added to highlight this.

p.17 l.18: "We have used the radiosonde data from the . . . "
Here the radiosonde data is explained in detail, however this information should be stated earlier since many of these details are relevant for the colocations presented as well.

p.22 l.4: "... altitude range 10-35 km ..."
It was previous stated the high altitude was 32 km.

p.22 l.5: "... accuracy in the stratosphere of 1-2 K"
Again this should be precision, also earlier and in the abstract the numbers quoted were 1-3 K.

p.22 l.11: "The upper limit of HRTP is defined mainly by the sampling frequency of the photometers ..."
This seems contrary to what was stated earlier, that the upper limit was set to 32 km.

───────────────────────────

---

## Referee Comment (RC2) · Anonymous Referee #2 · 23 Nov 2018

The authors present an improved retrieval algorithm for high-resolution temperature profiles (HRTP) based on bi-chromatic stellar scintillation measurements from the GO-MOS instrument onboard Envisat, i.e., based on two photometers at different wavelengths measuring the time delay between the signals of different wavelengths which is proportional to the refraction angle difference. The retrieval algorithm is described in detail from computing the refraction angle profile, its inversion to the refractivity index profile and to the retrieval of temperature profiles. The resulting exemplary high-resolution temperature profiles are presented for different conditions (vertical/oblique occultations, bright/non-bright stars). Validation with collocated temperature profiles from radiosondes shows good agreement. Furthermore, the use of these

high-resolution temperature profiles for gravity wave analysis is demonstrated.

The paper is well written and well structured. The retrieval algorithm is described in good detail. I just have one major comment and a range of minor comments. I recommend publishing the manuscript after revision taking into account the reviewer's recommendations and providing necessary clarifications. Please find the list of comments below.

Main comment:

It does not become fully clear which changes and improvements in the presented algorithm lead to the improvement of the high-resolution temperature profiles. In my understanding, the improvement stems from the optimization approach (Bayesian regularization) in the new method leading to main improvements at lower altitudes and for oblique occultations. It includes full covariance information instead of variance information only for optimal weighting of measurements and a priori. However, what is the effect of dropping the condition of cross-correlation coefficient <0.7 between the photometer signals as used in the old approach. What exactly are the changes compared to former algorithms or other available algorithms and what is the effect of these changes? I recommend including a discussion on this, maybe a short summarizing paragraph at the end of section 3. Also in the conclusions section this information should be included.

Minor comments:

Page 6, line 5 to 9: You jump right into this section by saying that the new retrieval starts at 32 km and afterwards explain why. But it does not get entirely clear. I find the explanation that you give in paragraph two of the summary section much clearer. I recommend starting in section 3.1 with a more general explanation on the main limiting factors of the HRTP retrieval (at upper and lower altitudes) along the explanation given in the summary. Also check that the altitude limits are stated consistently throughout the text.

P6, L7: "estimated using ECMWF data" and P12, L18: "ECMWF&MSIS" Please specifiy which ECMWF data (analyses, forecasts) and MSIS data you are using as a priori.

P8, L10: "This approximation is valid for large samples." Can you give a number or magnitude?

P12, L12: "The error due to horizontal gradients of the refractive index at right angles to the direction of light propagation has been estimated in (Healy, 2001; Sofieva et al., 2004); it is less than 1 % for altitudes." The sentences is unclear, please reformulate: "right angles" change to "perpendicular to" ". . . less than 1% for altitudes." Please state for which altitudes the error is less than 1%.

P14, L4-5: "These temperature profiles are collocated with high-resolution radiosonde data from the SPARC data center (http://www.sparc.sunysb.edu/html/5 hres.html)." Please include at this place the complete information on the radiosonde data and on the collocation criteria you are using for your comparison. You provide it later in the section (Page 17, line 17 to page 18, line 6) so you just need to move the paragraph to the beginning of this section.

P14, L24: ". . .previous HRTP validation results . . ." Please add a reference here.

Technical/editorial comments:

Please check consistent writing of "Sect.", "Section" and of "Figure or "Fig." throughout paper text.

Please check throughout the manuscript citations integrated in the text, should be written (e.g., at P3, L13/14): ". . .in Dalaudier et al. (2006) and Sofieva et al. (2009c). . . ")

P1, L18: "in in-orbital plane occultations" change to "for in-orbital plane occulations"

P1, 24: "analysis" change to "analyses" or "for the analysis of"

P2, L2: insert "instrument" after "(GOMOS)"

P2, L32: "For the stratosphere, it covers roughly a decade between 10 and 100 meters (of vertical scale)...". Suggest to rather use "a magnitude of 10 m to 100 m" instead of "decade ...".

P3, L4: "...to understand better... " change to "...to better understand..."

P3, L27: "Section 4" correct to "Section 5" (on gravity wave analysis).

P3, L30: It is unusual and there is no need to have a separate section on the paper structure. Please remove the section header. "1.3 The paper structure". Just make a separate paragraph at the end of section one explaining the contents of the paper. I suggest to merging the last sentence in section 1.2 with the first sentence in current section 1.3.

P4, L1-2: Remove the sentence "The information about the GOMOS HRTP dataset and data access is presented in Section 6."

P4, L2: "conclude the paper (Sect. 7)" correct to "conclude the paper in Section 6."

P6, figure2: There is no reference in the manuscript text to Figure 2.

P6, L6: "strength of scintillation" change to "the strength of scintillations"

P8, L9: "... where n is the size of samples participating in..." rather write "...where n is the sample size used in ..."

P8, Figure 4 (right): The thin light blue line and thin light red line are hardly visible in the plot. Please make it better visible and also mention them in the last sentence in the caption of Figure 4.

P9, L11: " produce scintillation during stellar occultation" Use plural ?  scintillations, occultations

P9, L24: "photometers" change to "photometer"

P10, Figure 5: Please make the green lines a bit thicker, especially in sub-panels B

and D.

P11, L7: "(7)" change to "(Eq.7)"

P14, L12: "raise" change to "rise"

P15, L15: "in other occultations" change to "for other occultations"

P14, L6-7: "The collocated temperature profiles are shown by blue lines in the left panels of Figs. 6 and 7, and the information about the spatio-temporal difference is provided in the figure." This sentence can be removed as the information is given in the figure caption.

P15, L6: change "Figure" to "figure title".

P20, L23: "of polar night jet" change to "of the polar night jet"

P20, L23: "...The enhancements in the equatorial region is also observed, which seem to be..." change to "...The enhancement in the equatorial region is also observed, which seems to be..."

P22, L7: "occultations bright stars" change to "occultations of bright stars"

P22, L20: "...constitute absolute majority ..." change to "...constitute the majority..."

---

## Author Comment (AC1) · 21 Dec 2018

Dear Reviewer,

Thank you very much for your attention and comments on our manuscript. Please find below our detailed replies on your comments.

Improvements could be made to better explain/quantify the effect of regularization on the time delay signal. I found it difficult to intuitively understand what the measurement fraction (Eq. 10) is saying in relation to other commonly used metrics of the effect of regularization. Since the time delay regularization approach is essential an optimal estimation style linear retrieval with K = I, would it instead be possible to calculate the equivalent averaging kernel and show rows/their sum? This would also give a sense of what effect the regularization has on the vertical resolution of the time delays.

Authors: We added a figure with examples of averaging kernels, for two occultations discussed earlier in the text: the oblique occultations R07673/S001, with strong influence of isotropic turbulence, and the vertical occultation R07588/S002, with small influence of turbulence. We added also a corresponding text discussing this figure.

A major point of the manuscript is the validation of small scale structure in the new HRTP FSP v1 data, and in particular its improvement over other processors. This is obviously a difficult task, but the validation efforts presented do show that the new data is more consistent with the radiosonde data than that of other processors. The part that I feel is missing is an explanation of why that improvement occurs. Differences in the regularization/upper limits are clearly outlined, however are these the only differences or are there others? What is the difference between the IPF and FMI v6 processing?

Other differences between IPF and FMI v6 processing are the retrieval grid, implementation of the Abel inversion and computing environment. The reason for excessive amplitude of fluctuations in HRTP produced by IPF v6 is unknown (we can only suspect that the reasons might be numeric instability or algorithm implementation), and we would prefer do not discuss this in the paper.

The manuscript would be improved by a brief overview of how the presented retrieval method is different than the IPF v6 and FMI v6 processing and how these differences can explain the observed improvement. If the primary reason for the improvement is the different regularization scheme then I think the effect of regularization needs to be better quantified, perhaps with the averaging kernel suggestion above.

Yes, the main change is the regularization. We added the figure with averaging kernel and corresponding discussion, as suggested.

p.1 l.17: "The HRTP profiles are retrieved with . . . high accuracy of ~1-3 K". As far as I can tell there is no discussion of accuracy anywhere in the discussion, this statement seems to apply to the HRTP precision. There are several other places in the abstract where accuracy should read precision.

Authors: We changed "accuracy" to "precision" and explained that "precision" means "the random uncertainty"

 "Furthermore, since the signal recorded by a detector is intrinsically one-dimensional, the retrieved parameter (temperature or density) is also one-dimensional." I think this statement is slightly misleading, it should be made clear that this is an approximation being made (the next paragraph does in fact justify this approximation).

We added an explanation that it is roughly along the trajectory of the ray perigee point in the atmosphere.

p.2 l.26: "The ratio of buoyancy frequency N to the Coriolis parameter f, N/f, is typically larger than 100." The ratio of limb horizontal path length to vertical path length is also about 100, does this mean that large scale horizontal fluctuations are expected to have an effect on GOMOS measurements?

The horizontal path length for occultation measurements in the stratosphere is ~2000 km. The vertical scale probing by the instrument depends on the sampling frequency. For GOMOS photometers with 1 kHz sampling frequency, this is ~ 3.4 m. The large-scale horizontal fluctuations do not affect the star scintillations. In our paper, we consider vertical and moderately oblique occultations (for which $\tan(\beta)$ is smaller than anisotropy of air density irregularities).

p.3 l.20: "... HRTP processed by the FMI scientific processor (analogous to the ESA IPF v6 algorithm) . . ." This is the only statement I could find about the relationship between the ESA and FMI v6, however from Fig. 9 there appears to be large differences between the two.

We added more details on differences between ESA and FMI v6.

p.6 l.6: "In the previous retrievals, the upper altitude, where the HRTP processing starts, depends on the strength of scintillation and value of a priori time delay". An additional statement is needed here as to why this was changed to be fixed at 32 km. Presumably in the previous version of the retrieval altitudes where the time delay was expected to be less than the sampling rate of the photometers was not used. Was it a problem with the a priori data not being accurate? Or something else?

We rephrased the statement into: "The sampling rate of GOMOS photometers allows determination of time delay up to ~35-38 km. However, above 32 km uncertainty of retrievals is large (see also discussion below), therefore the upper limit is set to 32 km in the new retrievals".

Below we discuss that, in addition to small time delay and the influence of instrumental noise, the isotropic turbulence affects the retrievals in oblique occultations mostly in the altitude range 30 -40 km.

p.9 l.16: "These situations are handled through regularization applied to the time delay profile . . . " I initially read this and was confused. The word regularization in standard limb retrievals is used because there is not enough information to retrieve the (essentially continuous) target quantity, but here the time delay can be found exactly. That being said, the technique applied is a standard regularization technique for limb retrievals and has a simple interpretation as 'virtual measurements' added to the system. I am not sure if it would be better to use a different word

(nothing immediately comes to mind), but it should be made clear that this is not regularization based upon the standard mathematical definition.

From our point of view, this is rather a standard case when regularization is needed. Although some value of time delay can be computed in cases of low correlation, this value has a very large uncertainty.

p.11 l.6: "If both matrices Ca and Cmeas are chosen to be diagonal, the Bayesian approach coincides with optimal filtration". This is only the case if what is referred to as "optimal filtration" does not include the CCC < 0.7 criteria that was used in v6. In fact the effect of removing this condition might be more important than going to full covariance matrices, is there an estimate of which of these two things is the dominant difference?

Yes, you are right, the condition "CCC<0.7" has a dominating effect and therefore the choosing Ca and Cmeas matrices diagonal would not give the results identical to V6. In the revised version, we removed this statement. In addition, according to comments of Reviewer #2, we added more discussion on the differences with the V6 algorithm (In Section 3 and in the Summary).

p.12 l.5: "Since the refractive angle is proportional to the time delay, its uncertainty can be easily obtained by multiplication of the time delay uncertainty by the corresponding factor." I assume that the uncertainty in the spectra is negligible compared to uncertainty in the time delay?

Yes, this is true. In the revised version, we noted this: "The uncertainty associated with effective wavelength determination is negligible compared to time delay uncertainty".

p.12 l.13: "The error due to horizontal gradients of the refractive index at right angles to the direction of light propagation has been estimated . . . it is less than 1% for altitudes" Is this for all altitudes or the altitudes relevant for the retrieval?

Sorry, the end of the sentence was missing: " for altitudes above 10 km"

p.12 l.18: "ECMWF & MSIS data is used at altitudes above HRTP range in the processing". Is there an estimate for how much of an effect this has on the final retrieved temperature?

The upper limit initialization affects the upper part of the profiles, with error rapidly decreasing (nearly exponentially with the atmospheric scale height ~ 7 km) so that ~3-5 km range of upper altitudes is affected. This is the same uncertainty as occurred in processing of radio-occultation data. The uncertainty due to upper-limit initialization is included in our processing at the stage of computing temperature profile uncertainty (Eq. (18) of the original manuscript).

p.13 l.23: "The regularization on time delay, as well as other inversion steps from time delay to temperature profile, can slightly degrade the vertical resolution ..." The effects here that can be quantified should be, in particular the effect of regularization.

As mentioned above, the illustration of averaging kernels after the regularization is added. We added also a reference to a publication on Abel inversion.

p.14 l.3: ". . . bright stars in vertical occultations . . . " Vertical here means in orbital plane?

Yes, they are the occultations in vertical plane; this definition of vertical occultations is given already on page 3, line 23.

p.14 l.5: ". . . from the SPARC data center (http://www.sparc.sunysb.edu/html/hres.html)" Link did not work for me when I tried it, but it might have just been temporarily down.

Thank you for noting this. The new link is  https://www.sparc-climate.org/data-centre/data-access/us-radiosonde/

p.14 l.26: "The HRTP wavenumber spectra in Figures 6 and 7 have visible cut-offs corresponding to scales ~150-250 m" It is hard to see this from the figure, maybe something could be added to highlight this.

In Figures 6 and 7, we added a red vertical lines, which indicates the small-scale regions (for those spectra where this cut-off is visible), where HRTP resolution can affect the spectra.

p.17 l.18: "We have used the radiosonde data from the . . . " Here the radiosonde data is explained in detail, however this information should be stated earlier since many of these details are relevant for the colocations presented as well.

In the revised version, we moved the description of radiosonde profiles used for validation to the beginning of Section 4.

p.22 l.4: ". . . altitude range 10-35 km . . . " It was previous stated the high altitude was 32 km.

Corrected to "10-32 km".

p.22 l.5: ". . . accuracy in the stratosphere of 1-2 K" Again this should be precision, also earlier and in the abstract the numbers quoted were 1-3 K.

Corrected to "1-3 K".

p.22 l.11: "The upper limit of HRTP is defined mainly by the sampling frequency of the photometers . . . " This seems contrary to what was stated earlier, that the upper limit was set to 32 km.

Here we discuss the principles of HRTP retrievals, not only the practical application to GOMOS. In the revised version, we added "In general, .." in the beginning of this sentence.

---

## Author Comment (AC2) · 21 Dec 2018

Dear Reviewer,

Thank you very much for your attention and comments on our manuscript. Please find below our detailed replies on your comments.

Reviewer #2. Main comment:

It does not become fully clear which changes and improvements in the presented algorithm lead to the improvement of the high-resolution temperature profiles. In my understanding, the improvement stems from the optimization approach (Bayesian regularization) in the new method leading to main improvements at lower altitudes and for oblique occultations. It includes full covariance information instead of variance information only for optimal weighting of measurements and a priori. However, what is the effect of dropping the condition of cross-correlation coefficient <0.7 between the photometer signals as used in the old approach. What exactly are the changes compared to former algorithms or other available algorithms and what is the effect of these changes? I recommend including a discussion on this, maybe a short summarizing paragraph at the end of section 3. Also in the conclusions section this information should be included.

Authors

Yes, the main effect is the introduced statistical optimization (Bayesian regularization). The effect of dropping the condition CCC<0.7 is clearly seen in Figure 5. The regions with CCC<=0.7 are clear visible in the panel D, where the blue line has drops from 1. The main rationale of the V6 retrieval method was using minimum a priori information in retrievals.

During the development of the algorithm, we tested the dropping CCC<0.7 condition (and applying weighting at all altitudes, not only at layers with low correlation. This is the equivalent of Bayesian regularization with diagonal matrices). As expected, profiles were smoother than in V6. We found (also expectedly) that the best results are when covariance matrices have off-diagonal elements: it is also justified by the retrieval principle.

There have been only two previous versions of the HRTP algorithm. In the first HRTP algorithm (developed before the launch), the values with CCC<0.7 were replaced with ECMWF-estimated time delay. The jumps in temperature profiles were sharp and unrealistic. In V6, the values with CCC<0.7 are replaced with the weighted mean of measured and a priori time delay. The unrealistic jumps became smaller, but still HRTP fluctuations are too large for oblique occultations.

In the revised version, we added more details in Section 3 (including illustration of averaging kernels, as suggested by Reviewer #1). In the discussion section, we also included the information about the main changes with respect to V6 and their influence on retrievals.

Reviewer#2, Minor comments:

Page 6, line 5 to 9: You jump right into this section by saying that the new retrieval starts at 32 km and afterwards explain why. But it does not get entirely clear. I find the explanation that you give in paragraph two of the summary section much clearer. I recommend starting in section 3.1 with a more general explanation on the main limiting factors of the HRTP retrieval (at upper and lower altitudes) along the explanation given in the summary. Also check that the altitude limits are stated consistently throughout the text.

Authors: this was also the comment by Reviewer#1. We added a short explanation on the selection of the upper level equal to 32 km, as well as a note that this will be discussed in more detail below.

P6, L7: "estimated using ECMWF data" and P12, L18: "ECMWF&MSIS" Please specifiy which ECMWF data (analyses, forecasts) and MSIS data you are using as a priori.

In the revised version, we clarified that ECMWF analyses data are used and MSIS90 model (Hedin, 1991).

P8, L10: "This approximation is valid for large samples." Can you give a number or magnitude?

Like generally in statistics, "large" is considered when n>~100. If n<~50, the estimates based on Student's t-distribution are used.

P12, L12: "The error due to horizontal gradients of the refractive index at right angles to the direction of light propagation has been estimated in (Healy, 2001; Sofieva et al., 2004); it is less than 1 % for altitudes." The sentences is unclear, please reformulate: "right angles" change to "perpendicular to" ": : : less than 1% for altitudes." Please state for which altitudes the error is less than 1%.

Sorry, the end of the sentence was missing: " for altitudes above 10 km".

P14, L4-5: "These temperature profiles are collocated with high-resolution radiosonde data from the SPARC data center (http://www.sparc.sunysb.edu/html/5 hres.html)." Please include at this place the complete information on the radiosonde data and on the collocation criteria you are using for your comparison. You provide it later in the section (Page 17, line 17 to page 18, line 6) so you just need to move the paragraph to the beginning of this section.

Thank you for your suggestion. In the revised version, we moved the information about the radiosondes to the beginning of Section 4.

P14, L24: ": : :previous HRTP validation results : : :" Please add a reference here.

The reference (Sofieva et al., 2009c) is added

Reviewer#2 Technical/editorial comments:

Please check consistent writing of "Sect.", "Section" and of "Figure or "Fig." throughout paper text.

Checked.

Please check throughout the manuscript citations integrated in the text, should be written (e.g., at P3, L13/14): ": : :in Dalaudier et al. (2006) and Sofieva et al. (2009c): : :")

Corrected

P1, L18: "in in-orbital plane occultations" change to "for in-orbital plane occultations"

P1, 24: "analysis" change to "analyses" or "for the analysis of"

Corrected

P2, L2: insert "instrument" after "(GOMOS)"

Done

P2, L32: "For the stratosphere, it covers roughly a decade between 10 and 100 meters(of vertical scale): : :". Suggest to rather use "a magnitude of 10 m to 100 m" instead of "decade : : :".

Changed to: "For the stratosphere, it is roughly between 10 and 100 meters"

P3, L4: ": : :to understand better: : : " change to ": : :to better understand: : :"

Corrected

P3, L27: "Section 4" correct to "Section 5" (on gravity wave analysis).

This is the remark explaining why HRTP V6 data were not recommended for GW research, and the illustration is provided in Sect.4

P3, L30: It is unusual and there is no need to have a separate section on the paper structure. Please remove the section header. "1.3 The paper structure". Just make a separate paragraph at the end of section one explaining the contents of the paper. I suggest to merging the last sentence in section 1.2 with the first sentence in current section 1.3.

Corrected according to the suggestion.

P4, L1-2: Remove the sentence "The information about the GOMOS HRTP dataset and data access is presented in Section 6."

P4, L2: "conclude the paper (Sect. 7)" correct to "conclude the paper in Section 6."

Corrected

P6, figure2: There is no reference in the manuscript text to Figure 2.

The references are added. In the revised manuscript, this is Figure 1.

P6, L6: "strength of scintillation" change to "the strength of scintillations"

Done

P8, L9: ": : : where n is the size of samples participating in: : :" rather write ": : :where n is the sample size used in : : :"

Done

P8, Figure 4 (right): The thin light blue line and thin light red line are hardly visible in the plot. Please make it better visible and also mention them in the last sentence in the caption of Figure 4.

In the revised version, we use more distinct colors: black and grey. They are now mentioned in the caption of Figure 4.

P9, L11: " produce scintillation during stellar occultation" Use plural ? scintillations, occultations

P9, L24: "photometers" change to "photometer"

Corrected

P10, Figure 5: Please make the green lines a bit thicker, especially in sub-panels B and D.

We made the green lines ticker.

P11, L7: "(7)" change to "(Eq.7)"

P14, L12: "raise" change to "rise"

P15, L15: "in other occultations" change to "for other occultations"

Corrected

P14, L6-7: "The collocated temperature profiles are shown by blue lines in the left panels of Figs. 6 and 7, and the information about the spatio-temporal difference is provided in the figure." This sentence can be removed as the information is given in the figure caption.

We would prefer keeping this sentence, because in the beginning of the paragraph we note that HRTP are shown by red lines.

P15, L6: change "Figure" to "figure title".

P20, L23: "of polar night jet" change to "of the polar night jet"

P20, L23: ": : :The enhancements in the equatorial region is also observed, which seem to be: : :" change to ": : :The enhancement in the equatorial region is also observed, which seems to be: : :"

P22, L7: "occultations bright stars" change to "occultations of bright stars"

P22, L20: ": : :constitute absolute majority : : :" change to ": : :constitute the majority: : :"

Corrected